# Modulation of SREBP Expression and Fatty Acid Levels by Bacteria-Induced ER Stress Is Mediated by Hemocyanin in Penaeid Shrimp

**DOI:** 10.3390/md21030164

**Published:** 2023-02-28

**Authors:** Zishu Huang, Xiaoyu Zheng, Zeyan Chen, Zhihong Zheng, Defu Yao, Shen Yang, Yueling Zhang, Jude Juventus Aweya

**Affiliations:** 1Fujian Provincial Key Laboratory of Food Microbiology and Enzyme Engineering, College of Ocean Food and Biological Engineering, Jimei University, Xiamen 361021, China; 2Guangdong Provincial Key Laboratory of Marine Biotechnology, Institute of Marine Sciences, Shantou University, Shantou 515063, China

**Keywords:** penaeid shrimp, pathogenic bacteria, endoplasmic reticulum (ER) stress, hemocyanin, SREBP, fatty acids

## Abstract

Many environmental and pathogenic insults induce endoplasmic reticulum (ER) stress in animals, especially in aquatic ecosystems, where these factors are crucial for life. In penaeid shrimp, pathogens and environmental stressors induce hemocyanin expression, but the involvement of hemocyanin in ER stress response is unknown. We demonstrate that in response to pathogenic bacteria (*Vibrio parahaemolyticus* and *Streptococcus iniae*), hemocyanin, ER stress proteins (Bip, Xbp1s, and Chop), and sterol regulatory element binding protein (SREBP) are induced to alter fatty acid levels in *Penaeus vannamei*. Interestingly, hemocyanin interacts with ER stress proteins to modulate SREBP expression, while ER stress inhibition with 4-Phenylbutyric acid or hemocyanin knockdown attenuates the expression of ER stress proteins, SREBP, and fatty acid levels. Contrarily, hemocyanin knockdown followed by tunicamycin treatment (ER stress activator) increased their expression. Thus, hemocyanin mediates ER stress during pathogen challenge, which consequently modulates SREBP to regulate the expression of downstream lipogenic genes and fatty acid levels. Our findings reveal a novel mechanism employed by penaeid shrimp to counteract pathogen-induced ER stress.

## 1. Introduction

Changes in the aquatic ecosystem due to environmental pollution, climate change, and anthropogenic activities threaten the lives of most animals [1]. Many factors and contaminants, including ammonia, heavy metals, ocean acidification, and pathogenic microbes, are among the common environmental stress factors that affect the survival of aquatic organisms [2,3,4,5]. Exposure of organisms to these environmental factors can induce various physiological responses, including metabolic reprogramming, oxidative stress, endoplasmic reticulum (ER) stress, etc. [3,6,7], which seems to be a strategy employed by organisms to counteract the harmful effects of these factors. Nevertheless, these responses could also have consequential effects on the organisms, increasing their risk of other damaging factors, including susceptibility to infections. 

Although changes in the external environment could affect many cellular components and physiological processes, these changes are particularly impactful on the ER because it is involved in many cellular functions, including calcium storage, protein synthesis, protein folding and processing, and lipid metabolism [8,9]. Thus, changes in oxidative stress, environmental stress, hypoxia, and calcium balance, affect the dynamic homeostasis of the ER, resulting in the accumulation of unfolded or misfolded proteins to induce ER stress [10,11,12,13]. In response to ER stress, cells activate a cytoprotective signaling pathway, i.e., unfolded protein response (UPR), which reduces protein synthesis, induces the expression or secretion of molecular chaperones, and degrades the misfolded proteins to help in the recovery process [14]. However, these cellular responses could also induce other physiological responses, including inflammatory response, apoptosis, and metabolic abnormalities, as well as lipid metabolism [15,16]. 

The activation of UPR involves three major signaling pathways, i.e., protein kinase RNA-like endoplasmic reticulum kinase (PERK), inositol requiring kinase 1 alpha (IRE1α), and activating transcription factor 6 (ATF6), which have also been implicated in the regulation of lipid synthesis [17]. In mammals, knockout or chemical inhibition of the phosphorylation of eukaryotic initiation factor-2α (eif2α), a downstream transcription factor of PERK, decreases lipid synthesis [18,19]. Similarly, high fructose induces lipid accumulation in human hepatocellular carcinoma cells (HepG2 cells) due to the activation of the spliced form of X-Box binding protein 1 (Xbp1s), a downstream transcription factor of IRE1α in ER stress [20]. On the other hand, Xbp1s deletion in the liver could lead to decreased de novo lipids synthesis [21]. Besides, liver fat accumulation further enhances ER stress, disrupting the homeostasis of liver lipid metabolism, thus forming a vicious circle that makes ER stress the cause and consequence of hepatic steatosis [22].

The sterol regulatory element-binding proteins (SREBPs) are key transcription factors that regulate lipid metabolism by controlling the expression of lipogenic genes [23]. SREBP binds via its C-terminal to SREBP cleavage-activating protein (SCAP) anchored to the ER membrane when inactive. However, upon initiation of lipid synthesis, the SCAP–SREBP complex moves into coat protein complex II (COP II) coated vesicles and is transported to the Golgi apparatus [24,25], where SREBP is cleaved by site-1 and site-2 proteases (S1P and S2P), to release active SREBP into the cytoplasm, followed by its entry into the nucleus to bind to SRE sequences in the promoter region of lipogenic genes [24,26]. Although the exact mechanism of SREBP regulation is still unclear, many studies have reported the involvement of ER stress. For instance, under ER stress, binding-immunoglobulin protein (Bip), a Heat shock protein 70 (Hsp70) chaperone, dissociates from the SREBP–SCAP complex or induces endoplasmic reticulum-associated degradation (ERAD) to break down the insulin-induced gene (INSIG) protein, hence, allowing the transport of SREBP to the Golgi for cleavage [27,28]. Similarly, the transcription factor Xbp1s, which plays a vital role in UPR, is necessary for SREBPs transcription [21,29]. 

We previously revealed that the SREBP homolog in *Penaeus vannamei* (*Pv*SREBP) modulates the expression of genes involved in fatty acid metabolism and immune-related functions [30], as well as many unannotated genes [31]. Most importantly, *Pv*SREBP could regulate the expression of the multifunctional respiratory protein hemocyanin [32] to affect the expression of other lipogenic and immune-related genes through a putative positive feedback mechanism [30]. Besides the role of hemocyanin in respiration and immune response [32,33,34], it plays a vital role in shrimp response to environmental cues, including hypoxia, pathogen stress, oxidative stress, ammonia stress, and thermal stress [35,36,37,38,39,40]. Given that hemocyanin is involved in shrimp response to various stressors and our recent finding that a putative positive feedback relationship exists between hemocyanin and SREBP, we wondered whether hemocyanin modulates ER stress to affect SREBP expression and fatty acid levels in penaeid shrimp. Thus, the current study explored the mediatory role of hemocyanin in bacteria-induced ER stress and the consequence on SREBP expression and fatty acid levels. Our data show that hemocyanin is a mediator of bacteria-induced ER stress that plays a crucial role in SREBP expression via a regulatory axis, which we designated as the “hemocyanin-ER stress-SREBP axis”.

## 2. Results

### 2.1. Bacteria Pathogens Induce Hemocyanin, SREBP, and ER Stress Proteins in Shrimp

We began by examining whether Gram-negative and Gram-positive bacteria could simultaneously induce hemocyanin, SREBP, and ER stress proteins expression in penaeid shrimp hepatopancreas, as ER stress modulates SREBP expression in mammals [41], while our recent study revealed a putative positive feedback relationship between *P. vannamei* hemocyanin (*Pv*HMC) and *Pv*SREBP expression [30]. Indeed, both mRNA transcripts and protein levels of *Pv*HMC (Figure 1A,B) and *Pv*SREBP (Figure 1C,D) were significantly induced in shrimp hepatopancreas after challenge with *V. parahaemolyticus* (Gram-negative) and *S. iniae* (Gram-positive). Similarly, increased levels of ER stress proteins (Bip and Xbp1s) were induced by the two bacteria at both the mRNA and protein levels (Figure 1E,F), with a marked increase in Bip protein (*Pv*Bip) expression induced by *V. parahaemolyticus* at 72 h post-infection (Figure 1E). These data suggest a relationship exists between hemocyanin and ER stress response in penaeid shrimp due to the concomitant bacteria-induced expression of *Pv*HMC, *Pv*SREBP, *Pv*Bip, and *Pv*Xbp1s.

### 2.2. Hemocyanin Interacts with ER Stress Proteins

Having shown that bacterial pathogens could simultaneously induce *Pv*HMC, *Pv*SREBP, Bip, and Xbp1s expression, we examined whether *Pv*HMC interacts with ER stress proteins, given that our preliminary data (unpublished) from GST pull-down followed by LC-MS/MS analyses identified Bip as one of the proteins that potentially interact with *Pv*HMC in shrimp hepatopancreas. Using in vitro GST pull-down analysis with recombinant GST-PvBip and PvHMC proteins, we confirmed a direct interaction between *Pv*Bip and PvHMC (Figure 2A). Next, we examined the relationship between *Pv*HMC and ER stress in penaeid shrimp using in vitro RNAi-mediated knockdown. Successful knockdown of *Pv*HMC (Figure 2B) resulted in significant attenuation in the levels of both mRNA transcripts and proteins of *Pv*Bip (Figure 2C), *Pv*Xbp1s (Figure 2D), and *Pv*Chop (Figure 2E) compared with control. To further ascertain the role of *Pv*HMC in ER stress response, *Pv*HMC was overexpressed in *Drosophila* S2 cells (due to the absence of suitable shrimp cell lines), and the effect on *Drosophila* Bip and Xbp1s proteins expression was examined. As expected, cells overexpressing *Pv*HMC also expressed significantly high protein levels of Bip (Figure 2F (upper blot) and Xbp1s (second blot)) at 36 h and 48 h post-transfection compared with control, albeit the expression of Bip protein was relatively higher than Xbp1s. These data indicate that *Pv*HMC could modulate ER stress by interacting with Bip and Xbp1s.

### 2.3. Bacteria-induced ER Stress Promotes SREBP Expression to Modulate Fatty Acid Levels in Shrimp

Although ER stress could regulate SREBP-mediated lipid metabolism in vertebrates [29,42], the existence of such a phenomenon in penaeid shrimp is unknown. Thus, having shown that *V. parahaemolyticus* induced high levels of *Pv*SREBP mRNA transcripts and protein levels after *V. parahaemolyticus* infection (Figure 1C), we examined the consequence on ER stress proteins. Besides the significant increases in mRNA transcripts and protein levels of *Pv*Bip (Figure 3A), PvXbp1s (Figure 3B), and PvChop (Figure 3C), there were significant changes in the profiles of various fatty acids in shrimp hepatopancreas after *V. parahaemolyticus* challenge compared with control (Table 1). Notably, a marked increase in the levels of arachidonic acid (ARA), eicosapentaenoic acid (EPA), and docosapentaenoic acid (DPA) was observed (Table 1).

### 2.4. ER Stress Activation or Inhibition Modulates SREBP and Fatty Acid Levels in Shrimp

To further ascertain the role of ER stress in promoting fatty acid metabolism in penaeid shrimp, as in other species, shrimp were treated with an ER stress activator, tunicamycin (designated TM), or inhibitor 4-Phenylbutyric acid (designated 4-PBA). The mRNA transcripts and protein levels of *Pv*Bip (Figure 4A), *Pv*Xbp1s (Figure 4B), and *Pv*Chop (Figure 4C) were all significantly elevated in shrimp hepatopancreas upon treatment with TM, albeit to different degrees. Moreover, TM treatment induced a significant increase in the mRNA transcripts and protein level of *Pv*SREBP (Figure 4D), resulting in an upregulation in the expression of downstream fatty acid metabolism-related genes, i.e., *PvFABP* (Figure 4E) and *PvCOX* (Figure 4F). On the other hand, treatment of shrimp with the ER stress inhibitor, 4-PBA, resulted in significant attenuation in the mRNA transcripts and protein levels of *Pv*Bip (Figure 4G), *Pv*Xbp1s (Figure 4H), and *Pv*Chop (Figure 4I), as well as *Pv*SREBP (Figure 4J) compared with control. A significant decrease in the mRNA transcripts of *PvFABP* (Figure 4K) and *PvCOX* (Figure 4L), downstream fatty acids metabolism-related genes, was also observed upon 4-PBA treatment. As expected, 4-PBA and TM treatment also changed the hepatopancreas lipids profile, as triglyceride levels (Figure 4M) and several crucial polyunsaturated fatty acids, such as DPA and DHA, were significantly decreased after 4-PBA treatment (Table 2), while treatment with TM induced elevated levels of triglycerides (Figure 4N) and various fatty acids, especially ARA and EPA in shrimp hepatopancreas compared with control (Table 3). These results suggest some form of the “ER stress-SREBP-lipid metabolism” axis in penaeid shrimp.

### 2.5. Hemocyanin Modulates ER Stress Proteins to Regulate SREBP Expression

Since our preliminary unpublished results show that hemocyanin (*Pv*HMC) interacts with ER stress proteins in the hepatopancreas, we wondered whether *Pv*HMC modulates ER stress to affect SREBP expression and lipid metabolism. Using RNAi-mediated knockdown of *Pv*HMC, we found that upon *Pv*HMC silencing (Figure 5A), the mRNA transcripts and protein levels of *Pv*Bip (Figure 5B), *Pv*Xbp1s (Figure 5C), *Pv*Chop (Figure 5D), and *Pv*SREBP (Figure 5E) were all significantly attenuated. Moreover, the mRNA transcripts of downstream fatty acid metabolism-related genes, i.e., *PvFABP* (Figure 5F) and *PvCOX* (Figure 5G), were also significantly decreased after *Pv*HMC knockdown. These results indicate that *Pv*HMC modulates ER stress in penaeid shrimp to regulate *Pv*SREBP expression and, consequently, fatty acid metabolism. It also further supports our contention that some form of “hemocyanin-ER stress-SREBP-lipid metabolism” exists in penaeid shrimp.

To ascertain the existence of this “hemocyanin-ER stress-SREBP-lipid metabolism” axis in shrimp, we used RNAi-mediated knockdown of *Pv*HMC followed by ER stress activator (TM) treatment. The protein levels of *Pv*Bip, *Pv*Xbp1s, *Pv*Chop, and *Pv*SREBP in *Pv*HMC knockdown samples treated with diluted DMSO (ds*Pv*HMC + DMSO) were significantly decreased compared with dsEGFP (control) samples treated with DMSO (dsEGFP + DMSO) (Figure 5H (lanes 1 and 3)). On the other hand, *Pv*HMC knockdown followed by TM treatment (ds*Pv*HMC TM) resulted in a significant increase in the protein levels of *Pv*Bip, *Pv*Xbp1s, *Pv*Chop, and *Pv*SREBP, compared with *Pv*HMC knockdown samples treated with diluted DMSO (ds*Pv*HM + DMSO) (Figure 5H (lanes 2 and 4)). These results indicate that *Pv*HMC regulates *Pv*SREBP expression through ER stress via the proposed “hemocyanin-ER stress-SREBP-lipid metabolism” axis. 

### 2.6. Hemocyanin-Mediated ER Stress Alters Fatty Acid Profile in Shrimp Hepatopancreas

Having observed that *Pv*HMC modulates ER stress proteins in the hepatopancreas, we explored the consequence of *Pv*HMC knockdown with or without ER stress activator (TM) treatment on fatty acid metabolism. We observed that *Pv*HMC knockdown without TM treatment (ds*Pv*HMC + DMSO) resulted in significant attenuation in the mRNA transcripts of *Pv*COX and *Pv*FABP, whereas *Pv*HMC knockdown followed by TM treatment (ds*Pv*HMC + TM), significantly increased the mRNA transcript levels of *PvCOX* (Figure 6A) and *PvFABP* (Figure 6B) compared with control. Besides, the ER stress activator (TM) significantly induced the expression of *PvCOX* and *PvFABP* in treated control samples (dsEGFP + TM) compared with untreated control samples (dsEGFP + DMSO). Analysis of the fatty acid profile in shrimp hepatopancreas further revealed that *Pv*HMC knockdown followed by TM treatment (ds*Pv*HMC + TM) significantly increased the levels of various fatty acids, especially arachidonic acid (C20:4n−6), eicosapentaenoic acid (C20:5n−3), and docosahexaenoic acid (C22:6n−3) (Table 4). On the other hand, *Pv*HMC knockdown followed by diluted DMSO (vehicle) treatment (ds*Pv*HMC + DMSO) resulted in decreased levels of most fatty acids compared with control, dsEGFP + DMSO (Table 5). These results further illustrate that *Pv*HMC modulates ER stress proteins to regulate *Pv*SREBP and fatty acid levels in penaeid shrimp.

## 3. Discussion

In this study, we establish that bacteria induce ER stress mediated by the multifunctional protein hemocyanin to modulate SREBP expression and fatty acid levels in penaeid shrimp. This regulatory mechanism is via an axis we designate as the “hemocyanin-ER stress-SREBP axis”. The current study’s revelation of a mediatory role of hemocyanin in ER stress provides the first mechanistic insight into the regulation of SREBP expression by hemocyanin to modulate fatty acid profile in response to pathogen-induced ER stress in crustaceans. These findings further confirm our previously proposed model of a positive feedback relationship between hemocyanin and SREBP expression in penaeid shrimp [30].

The hepatopancreas in Decapod crustaceans mainly integrates immune and metabolic processes [43], which is why it is involved in many physiological and pathophysiological functions, such as the production of digestive enzymes, absorption of digested food products, lipid, and carbohydrates metabolism, etc. [44]. In penaeid shrimp, the hepatopancreas expresses the highest mRNA transcripts of hemocyanin [43,45], which could be induced by environmental stress factors [35,39] and pathogens [46]. Here, we find that bacteria pathogens induced high levels of hemocyanin in shrimp hepatopancreas, coupled with a concomitant increase in the mRNA transcripts and protein levels of SREBP and key ER stress-related proteins (Figure 1). We contend that a regulatory relationship exists between hemocyanin and ER stress proteins in the hepatopancreas, given that hemocyanin is a multifunctional immune response protein [38]. Indeed, we find that hemocyanin (*Pv*HMC) interacts with ER stress proteins in protein–protein interaction studies, especially with Bip. Moreover, while *Pv*HMC knockdown attenuated the expression levels of Bip, Xbp1s, and Chop, *Pv*HMC overexpression increased their expression (Figure 2). 

As a molecular chaperone involved in de novo protein synthesis, Bip binds with misfolded proteins to induce their degradation [47]. Moreover, Bip’s interaction with other proteins can often trigger ER stress. An interaction between Bip and p53 is induced, for instance, when bacterial lipopolysaccharides (LPS) stimulate ER stress and intestinal dysfunction [48]. Similarly, amyloidogenic lysozyme variants can bind with Bip and accumulate in the ER to induce ER stress that promotes cell apoptosis [49]. It is, therefore, conceivable that bacteria pathogens induce hemocyanin expression and its interaction with Bip to modulate ER stress in shrimp hepatopancreas to regulate other cellular functions.

The accumulation of misfolded or unfolded proteins in the ER leads to ER stress, followed by unfolded protein response (UPR), an intracellular response [11,12,13] activated via the PERK, IRE1α, and ATF6 signaling pathways [50]. Given the crucial role of the ER in cellular protein quality control and homeostasis, ER stress can affect many cellular processes, including energy production, lipid metabolism, etc. [16,51,52] and several pathophysiological conditions [50,53,54]. Here, we find that hemocyanin, ER stress proteins (Bip, Xbp1s, and Chop), SREBP, and fatty acid metabolism-related genes (FABP and COX) are all induced by bacteria (*V. parahemolyticus*), together with changes in the fatty acid profile in the hepatopancreas (Figure 3 and Table 2). The three main signaling pathways (i.e., PERK, IRE1α, and ATF6) that modulate ER stress and UPR also play significant roles in lipid metabolism, with the transcription factor Xbp1s considered the master regulator of these signaling pathways [55], and therefore crucial in lipid metabolism [17,20,21]. 

In lipid metabolism, SREBP is a crucial transcription factor that regulates lipid homeostasis and has also been implicated in ER stress-induced lipid metabolism disorders [56,57]. Here, when ER stress was inhibited with 4-Phenylbutyric acid (4-PBA), the expression levels of Bip, Xbp1s, and Chop, together with SREBP, FABP, COX, triglycerides, and several fatty acids are all attenuated. However, their levels increased when treated with the ER stress activator tunicamycin (TM) (Figure 4, Table 3 and Table 4). Although the relationship between Xbp1s and SREBP in ER stress-induced lipid metabolism is unknown, with no clear understanding of the mechanisms involved, Xbp1s is reported to play a role in SREBP activation during ER stress [17,58] through direct or indirect regulation [21,59], or by affecting SREBP activity via the transcriptional regulation of oxysterol binding proteins (OSBPs) [60]. Similarly, mice hepatic Xbp1s is necessary for the translation of SREBP (SREBP1c) and, therefore, its targets [29]. Therefore, given that hemocyanin (*Pv*HMC) knockdown attenuated the expression of Bip, Xbp1s, Chop, and SREBP, whereas *Pv*HMC overexpression increased their expression at both the transcriptional and protein levels (Figure 2), we contend that *Pv*HMC plays a mediatory role in ER stress response to modulate SREBP expression and fatty acid metabolism. Indeed, our recent study revealed a putative positive feedback relationship between hemocyanin and SREBP in *P. vannamei* [30], although the significance of this relationship was not ascertained. Nonetheless, given that in mammals, ER stress could regulate SREBP expression [29,55], while our current data show the bacteria-induced expression of hemocyanin, SREBP, and ER stress proteins (Bip, Xbp1s, and Chop), and hemocyanin knockdown or overexpression affected the expression levels of Bip, Xbp1s, Chop, SREBP, and fatty acid metabolism; it is, therefore, conceivable to state that hemocyanin modulates ER stress to regulate SREBP, its downstream genes, and lipid metabolism through an immunometabolism regulatory axis, which we designated as the “hemocyanin-ER stress-SREBP axis”.

In addition to being a key transcription factor in the ER stress pathway that regulates liver-related fatty acid synthesis [21], Xbp1s also regulates genes involved in various cellular processes [55,61]. Thus, the attenuation of ER stress proteins (Bip, Xbp1s, and Chop) expression in shrimp hepatopancreas after hemocyanin knockdown indicates that hemocyanin plays an essential mediatory role in ER stress, especially during pathogen challenge. The ER stress mediatory role of hemocyanin could probably be a protective mechanism used by penaeid shrimp to counteract pathogen-induced stress on other pathophysiological processes, such as immune response, given that hemocyanin is an immune response protein [46]. The ability of hemocyanin to regulate lipid synthesis through ER stress was particularly intriguing, given that some of the dysregulated fatty acids, especially the polyunsaturated fatty acids, such as arachidonic acid (ARA), eicosapentaenoic acid (EPA), and docosahexaenoic acid (DHA) (Table 2, Table 3 and Table 4) have antibacterial, antiviral and immunomodulatory effects [62,63]. Therefore, under pathological infection conditions, hemocyanin (an immune response protein) activates the expression of SREBP through ER stress, which regulates the expression of lipogenic genes and fatty acid levels (Figure 7), forming an immune metabolic network that counteracts the ER stress and augments the immune response. However, this immune-metabolic network model requires further research to substantiate.

In summary, our data demonstrate that pathogenic bacteria infection induces hemocyanin expression to modulate ER stress, thereby regulating SREBP expression and lipid metabolism in shrimp hepatopancreas. These results indicate that pathogen-induced ER stress, mediated by hemocyanin, could be a novel molecular mechanism that links immunity and lipid metabolism through an immunometabolism regulatory network in shrimp hepatopancreas.

## 4. Materials and Methods

### 4.1. Experimental Animals

Healthy *Penaeus vannamei* (10 ± 2 g in size) purchased from Shantou Huaxun aquatic products Co., Ltd., Shantou, Guangdong, China (23°21′14.73″ N and 116°40′55.1″ E), were cultured in laboratory tanks containing circulating seawater (24 ℃ and 0.5% salinity) for 2–3 days acclimatization. Shrimp were fed twice daily with commercial feed, and only active shrimp without signs of diseases or spots on their bodies were used for the experiments. Although, no ethical approval is required for experimental work with crustaceans in China (Regulations of Guangdong Province on the Administration of Experimental Animals (http://www.gd.gov.cn/zwgk/wjk/zcfgk/content/post_2524545.html accessed on 25 May 2022) and Regulations of the People’s Republic of China on the Administration of Experimental Animals (https://kyc.jnmc.edu.cn/2021/0826/c2735a122933/page.htm accessed on 25 May 2022), all animal experiments complied with the institutional guidelines of the animal research and ethics committee of Shantou University, Guangdong, China.

### 4.2. Reagents

Phenylmethylsulfonyl fluoride (PMSF) (Cat# ST505) was purchased from Beyotime Biotechnology, Shanghai, China. Lysis buffer containing protease inhibitor (Cat# CW233) was obtained from Cwbio, Beijing, China. The 4-Phenylbutyric acid (Cat# HY-A0281) was purchased from MCE, New Jersey, USA, while tunicamycin (CAS# 11089-65-9) was from Sangon Biotech, Shanghai, China. Dimethyl sulfoxide (CAS#67-68-5) was purchased from MP Biomedicals, Irvine, CA, USA.

### 4.3. Challenge Experiments and Samples Processing

In the pathogen-challenged experiments, pre-acclimatized healthy shrimp, divided into three groups (50 shrimp each), were each injected intramuscularly with 100 μL of *Vibrio parahaemolyticus* MCCC 1A02609 (1 × 10^5^ CFU/mL) or *Streptococcus iniae* (GenBank: NZ_JH930418.1) (1 × 10^5^ CFU/mL). Control group shrimp were injected with an equal volume of sterile phosphate-buffered saline (PBS). In the ER stress inhibitor or activator treatment experiments, three groups (10 Shrimp each) were injected with 100 μL (0.1 µg/µL) of 4-Phenylbutyric acid (4-PBA) or with 100 μL (6 ng/µL) of tunicamycin (TM). The control group shrimp were each injected with an equal volume of 6% diluted DMSO (dimethylsulfoxide). At different time points (i.e., 0, 25, 48, and 72 h) post-pathogen injection, six randomly selected shrimp per group were anesthetized on ice, followed by removing their hepatopancreas, one-half of which was used for immediate RNA extraction or snap-frozen in liquid nitrogen and stored at 80 °C for later use. The other one-half was processed for SDS-PAGE and Western blot analysis as previously described [30]. The processed samples were used immediately or stored at −20 °C for later use. Hepatopancreas samples from shrimp treated with the ER stress inhibitor and activator were collected at 6 h post-injection and processed as above for RNA extraction and cell lysis.

### 4.4. Total RNA Extraction, cDNA Synthesis, and Quantitative RT-PCR

Total RNA was extracted from hepatopancreas samples using the RNA Fast 200 kit (Cat# 220011, FeiJie, Shanghai, China). RNA concentration was determined with a NanoDrop 2000 spectrophotometer (Model# ND-ONE-W, Nano-drop Technologies, Wilmington, DE, USA), and the purity was ascertained by the 260/280 ratio (≥2.0) and 1% agarose gel electrophoresis. Only high-quality total RNA samples were used for cDNA synthesis with the TransScript^TM^ One-step gDNA removal and cDNA Synthesis SuperMix kit (Cat# AT311, TransGen Biotech, Beijing, China). The cDNA samples were used immediately or stored in aliquots at −20 °C. Quantitative polymerase chain reaction (qPCR) analyses were carried out with the qTOWER 3G Real-Time PCR system (Model# 1016-67, Analytik Jena AG, Germany). Each qPCR reaction contained 10 μL of 2× RealStar Green power mixture (Cat#A311-10, GenStar, Beijing, China), 1 μL cDNA template (10 ng/μL), 1 μL each of forward and reverse primers (10 μmol), and 7 μL ddH2O. The following qPCR cycling conditions used were as follows: one cycle at 95 °C for 10 min and 40 cycles at 95 °C for 15 s and 60 °C for 30 s. The relative gene expression was calculated by the 2^−ΔΔCT^ method [64] with the *EF1α* gene of *P. vannamei* (*PvEF1α*) used as the housekeeping gene. Triplicate samples were analyzed per treatment and repeated for at least three biological samples. The PCR primer sequences are shown in Table 5.

**Table 5 marinedrugs-21-00164-t005:** Primers used in this study.

Primers	Sequence (5′-3′)	Amplicon Size (bp)
	**Primers for protein expression**	
*Pv*Bip-F	ATGAGGTGTTGGACTGCA	1926
*Pv*Bip-R	CTACAATTCGTCCTTTTCATA	
pGEX-6P-1-*Pv*Bip-F	TCCAGGGGCCCCTGGGATCCATGAGGTGTTGGACTGCATTAG	1950
pGEX-6P-1-*Pv*Bip-R	CCCGGGAATTCCGGGGATCCCAATTCGTCCTTTTCATAATCT	
	**Primers for Real-time PCR**	
*Pv*HMC-qF	CCTGGCCTCATAAAGACAACA	104
*Pv*HMC-qR	TTTTCCACCCTTCAAAGATACC	
*Pv*EF-1α-qF	TATGCTCCTTTTGGACGTTTTGC	118
*Pv*EF-1α-qR	CCTTTTCTGCGGCCTTGGTAG	
*Pv*Bip-qF	GAGCGTCTGATTGGTGATT	163
*Pv*Bip-qR	GTGGCTTGTCGTTCTTGTT	
*Pv*Xbp1s-qF	AACTACGGGACCTGACATCTGC	207
*Pv*Xbp1s-qR	ACTGCCTTCTGCTGATCCACC	
*Pv*Chop-qF	TGACCCCCACCACCATCCC	204
*Pv*Chop-qR	ACTCGCTCCTCCGTCTCCC	
*Pv*SREBP-qF	GGAGTTGTTGTTGCCGTGG	134
*Pv*SREBP-qR	TGGCTGAGATGTTGGTAATGG	
*Pv*FABP-qF	CGCTAAGCCCGTGCTGGAAGT	103
*Pv*FABP-qR	CTCCTCGCCGAGCTTGATGGT	
*Pv*COX-qF	CCACAAGCGACTGATGACTTA	103
*Pv*COX-qR	GTAGGCATTGAGGGTGATGTAG	
	**Primers for dsRNA**	
ds*Pv*HMC-T7F	GGATCCTAATACGACTCACTATAGGGTCCTCATCCACTGCAAA	449
ds*Pv*HMC-R	TTGGACAGACGTTCAGCA	
ds*Pv*HMC-T7R	GGATCCTAATACGACTCACTATAGGTTGGACAGACGTTCAGCA	449
ds*Pv*HMC-F	GTCCTCATCCACTGCAAA	
dsEGFP-T7F	GGATCCTAATACGACTCACTATAGGCGTAAACGGCCACAAGTT	429
dsEGFP-R	TTCACCTTGATGCCGTTC	
dsEGFP-T7R	GGATCCTAATACGACTCACTATAGGTTCACCTTGATGCCGTTC	429
dsEGFP-F	CGTAAACGGCCACAAGTT	

### 4.5. Targeted RNA Interference (RNAi)

Double-stranded RNA (dsRNA) targeting the hemocyanin gene of *P. vannamei* (GenBank: X82502.1) designated dsPvHMC or targeting the gene encoding enhanced green fluorescent protein (GenBank: UDY80669) designated dsEGFP were synthesized using the T7 RiboMAX^TM^ Express RNAi System kit (Cat# P1700, Promega, Madison, WI, USA). The primer sequences used for the in vitro dsRNA synthesis are shown in Table 5. For the RNAi experiments, pre-acclimatized shrimp (10 shrimp per group) were injected intramuscularly with 100 μL (1 μg/g shrimp) of dsPvHMC or 100 μL (1 μg/g shrimp) of dsEGFP. At 48 h post-dsRNA injection, six randomly selected shrimp per group were anesthetized on ice before removing their hepatopancreas for total RNA extraction and cell lysate preparation as described above.

### 4.6. Plasmids

The cDNA corresponding to the open reading frame (ORF) of the *P. vannamei* hemocyanin gene (*PvHMC*, GenBank: X82502.1), obtained by PCR, was subcloned into the pIZ-V5/His vector (Cat# V8000-01, Invitrogen, Carlsbad, CA, USA) at the Xho restriction enzyme sites to generate pIZ-PvHMC-Flag. Similarly, the cDNA sequence of the ORF (1968 bp) of *P. vannamei Bip* gene (GenBank: ROT66321.1) was subcloned into the pGEX-6p-1 vector (Cat# 27-4597-01, GE Healthcare, Boston, MA, USA) to generate a pGEX-6p-1-PvBip construct with a GST tag and used to express the GST-PvBip recombinant protein. All the primer sequences used to amplify the DNA fragments are in Table 5.

### 4.7. Cell Culture and Transfection

*Drosophila Schneider* S2 cells, kindly provided by Prof. Jianguo He of Sun Yat-sen University, Guangzhou, China, were cultured at 27 °C for 48 h in Schneider’s Drosophila Medium (Cat# 21720-024, Invitrogen, Carlsbad, CA, USA) containing 10% fetal bovine serum (Cat# 10091148, Gibco Life Technologies, Grand Island, NY, USA). Confluent cells were plated onto 6-well plates (Cat#712001, NEST Biotechnology, Shanghai, China) at 1.5 × 10^6^ cells per well in 2 ml medium. At 60–80% confluence, cells were transfected with 2 μg of pIZ-PvHMC-Flag or the empty vector pIZ-V5/His using the FuGENE^@^ HD Transfection Reagent (Cat#E2311, Promega, Madison, WI, USA). At 36 h and 48 h post-transfection, cells were harvested, washed three times with pre-cooled PBS, and lysed with lysis buffer containing protease inhibitor (Cat# CW233, Cwbio, Beijing, China) plus 2× PMSF before being centrifuged at 20,000× *g* (4 °C, for 20 min). The obtained supernatant was mixed with 5× loading buffer and boiled for 10 min. Samples were used immediately or stored at −20 °C for later use.

### 4.8. GST Pull-Down, SDS-PAGE, and Western Blot Analyses

For GST pull-down analysis, purified recombinant proteins (i.e., GST-PvBip, PvHMC, and GST) were incubated with Glutathione Sepharose 4B beads (Cat#17-0756, GE Healthcare, Boston, MA, USA) at 4 °C for 2 h. Next, samples were washed ten times with 0.01 M PBS (containing 1% triton) before being boiled with 5× loading buffer for 10 min. For SDS-PAGE and Western blot analyses, the samples prepared and boiled with 5× loading buffer, as described above, were separated on SDS-PAGE before being transferred onto PVDF membranes (Cat# R0NB30936, Millipore, Billerica, MA, USA) using a Mini TransBlot cell wet transfer system (Model#1658030, Bio-Rad, Richmond, CA, USA). Next, membranes were blocked for 1 h at room temperature in 5% skimmed milk dissolved in TBST buffer (20 mM Tris, 150 mM NaCl, 0.1% Tween 20, pH 7.4) before being incubated overnight at 4 °C with the appropriate primary antibody followed by washing three times with TBST buffer (15 min each time). After incubation with the corresponding secondary antibody for 1 h at room temperature, membranes were washed three times with TBST buffer (15 min each time), followed by treatment with enhanced chemiluminescence (ECL) reagent (Cat# WBLUF0100, Millipore, Billerica, MA, USA). The signals were captured on Amersham Imager 600 (Model# V1.0.0, GE Healthcare, Boston, MA, USA).

### 4.9. Antibodies

The following antibodies and working concentrations were used: Mouse anti-tubulin antibody (Cat#T6074, Sigma-Aldrich, St Louis, MO, USA, 1:3000 for Western blot), Rabbit anti-Bip antibody (Cat#AB310, Beyotime, Shanghai, China, 1:1000 for Western blot), Rabbit anti-Xbp1s antibody (Cat#AF8366, Beyotime, Shanghai, China, 1:1000 for Western blot), Rabbit anti-Chop antibody (Cat#T6074, Affinity Biosciences, Cincinnati, OH, USA, 1:1000 for Western blot), Mouse anti-SREBP antibody (produced in-house, 1:1000 for Western blot), Rabbit anti-hemocyanin antibody (produced in-house, 1:3000 for Western blot), Rabbit anti-V5 antibody (Cat#D191104, Sangon Biotech, Shanghai, China, 1:1000 for Western blot), Mouse anti-FLAG antibody (Cat#D110005, Sangon Biotech, Shanghai, China, 1:1000 for Western blot), Mouse anti-GST antibody (Cat#HT601-01, 1:1000, TransGen Biotech, Beijing, China, 1:1000 for Western blot), Horseradish peroxidase (HRP)-linked goat anti-rabbit (Cat#31460, Thermo Fisher Scientific, Waltham, MA, USA, 1:3000, for Western blot), and HRP-linked goat anti-mouse (Cat#A32723, Thermo Fisher Scientific, Waltham, MA, USA, 1:3000 for Western blot). 

### 4.10. Fatty Acid Profiling and Triglycerides Analysis

The fatty acid profile of shrimp hepatopancreas was analyzed by gas chromatography (GC), as previously described [30]. Briefly, freeze-dried hepatopancreas samples mixed with chloroform: methanol (2:1) solution were incubated at 4 °C for 48 h before being filtered with 0.88% KCl (Cat#7447-40-7, XiLong Scientific, Shantou, China). Next, the filtered samples were dried with nitrogen, followed by the addition of KOH-methanol solution and 100 μL C17 standard (10 μg/μL) before being heated at 65 °C for 30 min. Boron trifluoride (Cat#134821, Sigma-Aldrich, St. Louis, MO, USA) was then added to each sample and heated at 70 °C for 15 min. Finally, saturated NaCl (Cat#10101101, XiLong Scientific, Shantou, China) and n-hexane (Cat#13700101, XiLong Scientific, Shantou, China) were added to each sample and centrifuged at 400× *g* (room temperature for 5 min) to collect the supernatants for analysis on the GC 2010-plus workstation (Model#2010, Shimadzu, Japan). Commercial standards (Cat#CRM47885, Sigma-Aldrich, St. Louis, MO, USA) were used to identify the fatty acids and their contents expressed as a percentage of total fatty acids.

The triglyceride levels in shrimp hepatopancreas were analyzed using a commercial triglyceride assay kit (Cat#A110-1-1, Jiancheng Bioengineering Institute, Nanjing, China). Briefly, 0.1g of hepatopancreas samples were immersed in absolute ethanol at a weight: volume ratio of 1:9, after which samples were centrifuged at 4500× *g* (4 °C for 10 min) to collect the supernatant. Samples treated with distilled water were used as the negative control, while those treated with the given standard were used as the positive control. Next, separate mixtures of 2.5 μL of distilled water (negative control), 2.5 μL of triglycerides standard (positive control), or 2.5 μL of hepatopancreas homogenate supernatant and 250 μL of the working solution were incubated at 37 °C for 10 min. The absorbance at 510 nm was then measured with a multi-mode microplate reader (BioTek, SynergyH1, Winooski, VT, USA), and the triglyceride content in each sample was calculated using the following formula:$$\mathrm{Triglyceride}\;(\mathrm{mmol}/L)=\frac{A_{\mathrm{sample}}-A_{\mathrm{control}}}{A_{\mathrm{standard}}-A_{\mathrm{control}}}\times C_{\mathrm{standard}}+\frac{W}{V_{\mathrm{extracting}\;\mathrm{solution}}}$$
where W is tissue weight (g), and V_extracting solution_ is the volume of absolute ethanol (L).

### 4.11. Statistical Analysis

Data are presented as mean ± SEM (standard error of the mean) from *n* = 3 biological replicates. Comparisons between groups were performed using Student’s *t*-test analysis, and significance was considered at *p* < 0.05 or less. 

## Figures and Tables

**Figure 1 marinedrugs-21-00164-f001:**
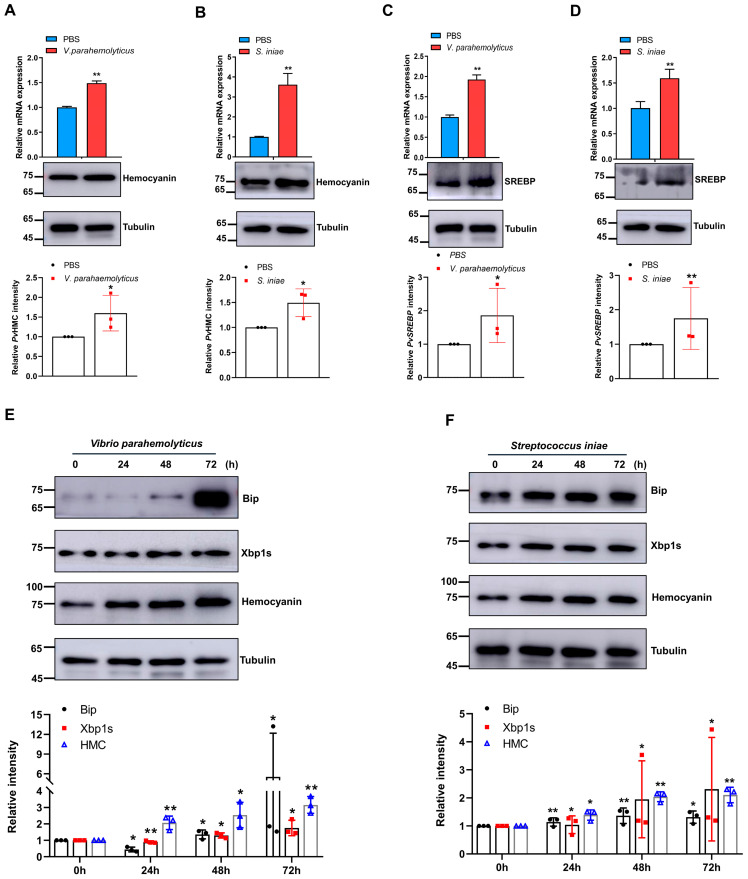
Bacteria pathogens induce the expression of hemocyanin, SREBP, and ER stress proteins in *Penaeus vannamei* hepatopancreas. *Pv*HMC mRNA and protein expression after challenge with (**A**) *Vibrio parahaemolyticus* and (**B**) *Streptococcus inia*e. *PvSREBP* mRNA and protein expression after challenge with (**C**) *V. parahaemolyticus*, and (**D**) *S. iniae* for 72 h. Protein expression levels of Bip, Xbp1s, and *Pv*HMC after challenge with (**E**) *V. parahaemolyticus*, and (**F**) *S. iniae* for 0, 25, 48, and 72 h. The mRNA levels were quantified by qRT-PCR and normalized to those of *EF1a* mRNA, while protein levels were determined by Western blot using the indicated antibodies, with tubulin used as the loading control. Protein band intensity was analyzed using ImageJ and normalized relative to tubulin. Results are reported as mean ± SEM (*n* = 3). * *p* < 0.05, ** *p* < 0.01 vs. control (PBS). The immunoblots shown are representative of at least three independent experiments. *Pv*HMC: hemocyanin; *Pv*SREBP: sterol regulatory element binding protein homolog of *Penaeus vannamei*; Bip: binding-immunoglobulin protein; Xbp1s: X-box-binding protein; PBS: phosphate-buffered saline.

**Figure 2 marinedrugs-21-00164-f002:**
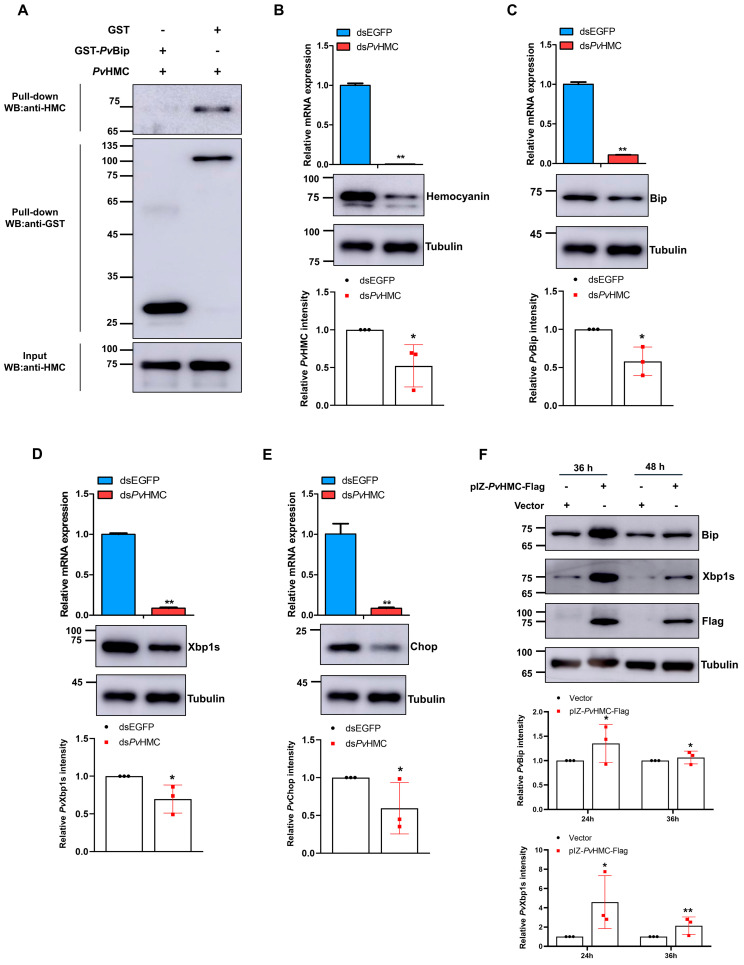
Hemocyanin interacts with and modulates ER stress proteins’ expression in *Penaeus vannamei* hepatopancreas. (**A**) Immunoblots of protein–protein interaction analysis between GST (control) and GST-*Pv*HMC or GST-*Pv*HMC and His-*Pv*Bip. The mRNA and protein expression levels of (**B**) *Pv*HMC, (**C**) *Pv*Bip, (**D**) *Pv*Xbp1s, and (**E**) *Pv*Chop after *Pv*HMC knockdown. (**F**) Bip and Xbp1s protein levels in *Drosophila* S2 cells overexpressing *Pv*HMC at the indicated time points. The mRNA levels were quantified by qRT-PCR and normalized to those of *EF1a* mRNA, while protein levels were determined by Western blot using the indicated antibodies, with tubulin used as the loading control. Protein band intensity was analyzed using ImageJ and normalized relative to tubulin. Results are reported as mean ± SEM (*n* = 3). * *p* < 0.05, ** *p* < 0.01 vs. control. The immunoblots shown are representative of at least three independent experiments. Chop: CCAAT-enhancer-binding protein homologous protein.

**Figure 3 marinedrugs-21-00164-f003:**
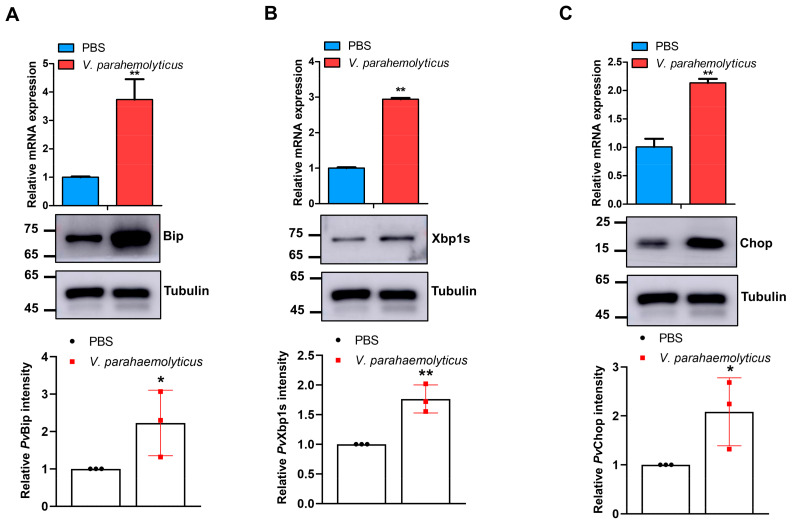
*Vibrio parahaemolyticus* induces the expression of ER stress proteins in *Penaeus vannamei* hepatopancreas. The mRNA and protein expression levels of (**A**) *Pv*Bip, (**B**) *Pv*Xbp1s, and (**C**) *Pv*Chop after challenge with *V. parahaemolyticus* for 48 h. The mRNA levels were quantified by qRT-PCR and normalized to those of *EF1a* mRNA, while protein levels were determined by Western blot using the indicated antibodies, with tubulin used as the loading control. Protein band intensity was analyzed using ImageJ and normalized relative to tubulin. Results are reported as mean ± SEM (*n* = 3). * *p* < 0.05, ** *p* < 0.01 vs. control. The immunoblots shown are representative of at least three independent experiments.

**Figure 4 marinedrugs-21-00164-f004:**
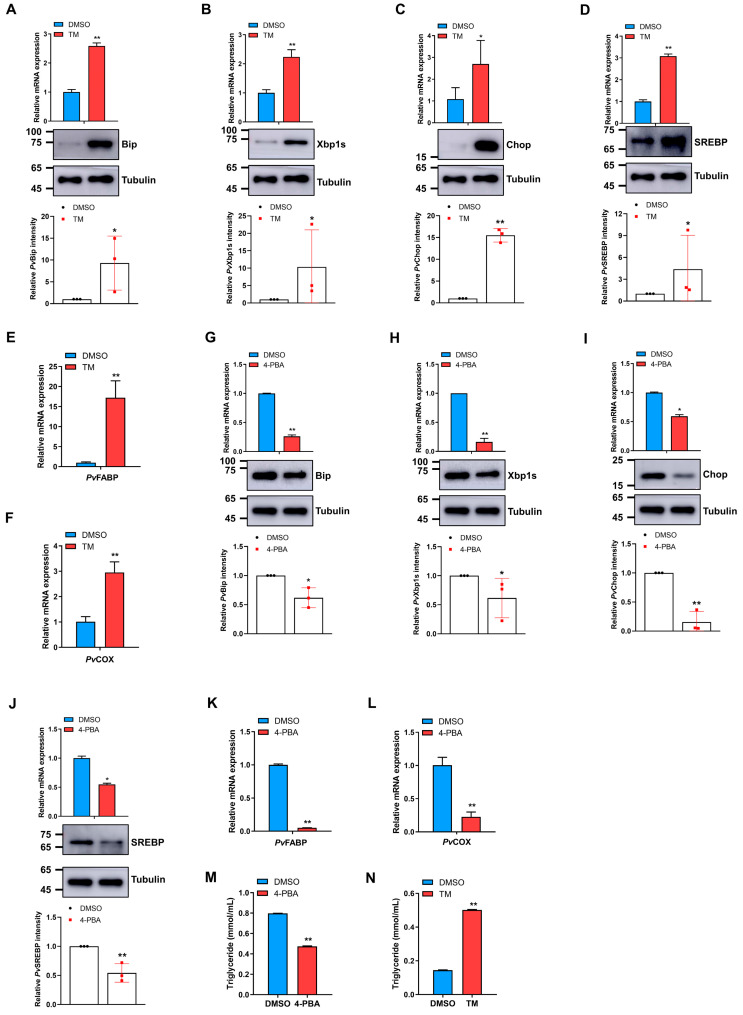
ER stress activator or inhibitor treatment affects the expression of ER stress proteins, SREBP, and fatty acid metabolism genes in shrimp hepatopancreas. The mRNA and protein expression levels of (**A**) *Pv*Bip, (**B**) *Pv*Xbp1s, (**C**) *Pv*Chop, and (**D**) *Pv*SREBP after treatment with ER stress activator tunicamycin (TM). mRNA levels of (**E**) *PvFABP* and (**F**) *PvCOX* after treatment with 6 ng/µL of tunicamycin (TM) for 6 h. mRNA and protein expression levels of (**G**) *Pv*Bip, (**H**) *Pv*Xbp1s, (**I**) *Pv*Chop, and (**J**) *Pv*SREBP after treatment with 0.1 µg/µL of 4-Phenylbutyric acid (4-PBA) for 12 h. mRNA levels of (**K**) *PvFABP* and (**L**) *PvCOX* after treatment with 0.1 µg/µL of 4-PBA for 12 h. Triglyceride levels after treatment with (**M**) 0.1 µg/µL of 4-PBA for 12 h and (**N**) 6 ng/µL of TM for 6 h. Control samples were treated with 6% diluted dimethyl sulfoxide (DMSO). The mRNA levels were quantified by qRT-PCR and normalized to those of *EF1a* mRNA, while protein levels were determined by Western blot using the indicated antibodies, with tubulin used as the loading control. Protein band intensity was analyzed using ImageJ and normalized relative to tubulin. Triglyceride levels were determined using a commercial kit. Results are reported as mean ± SEM (*n* = 3). * *p* < 0.05, ** *p* < 0.01 vs. control. The immunoblots shown are representative of at least three independent experiments. FABP: fatty acid binding protein, COX: cyclooxygenase.

**Figure 5 marinedrugs-21-00164-f005:**
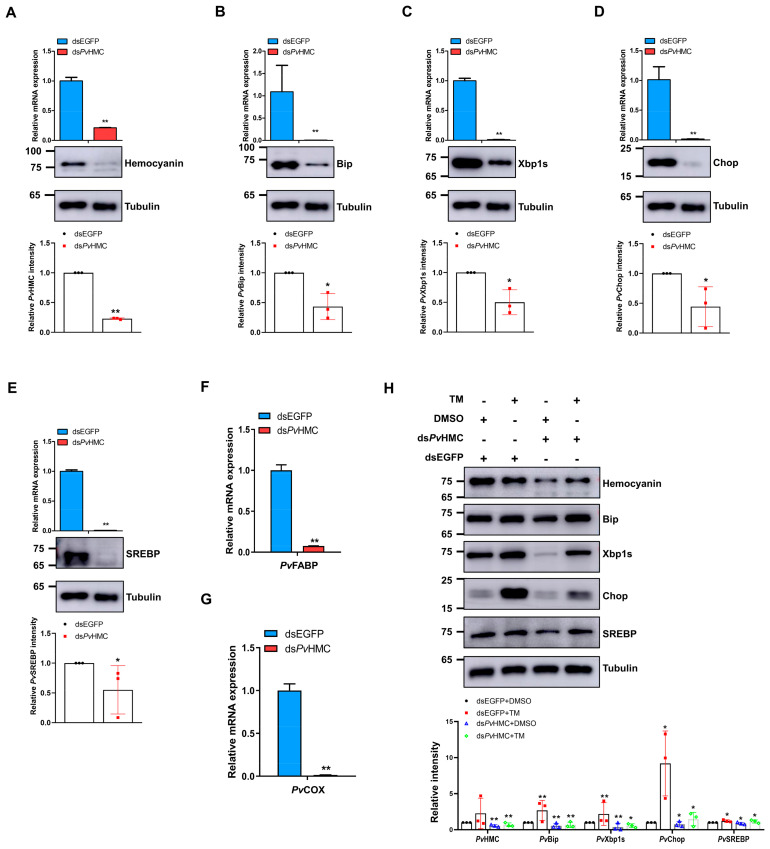
Hemocyanin modulates ER stress proteins to affect SREBP expression in *Penaeus vannamei* hepatopancreas. mRNA and protein expression levels of (**A**) *Pv*HMC, (**B**) *Pv*Bip, (**C**) *Pv*Xbp1s, (**D**) *Pv*Chop, and (**E**) *Pv*SREBP after *Pv*HMC knockdown. mRNA levels of (**F**) *PvFABP* and (**G**) *PvCOX* after *Pv*HMC knockdown. (**H**) Protein expression levels of *Pv*Bip, *Pv*Xbp1s, *Pv*Chop, and *Pv*SREBP after PvHMC knockdown followed by treatment with 6 ng/µL of tunicamycin (TM) or diluted dimethyl sulfoxide (DMSO) for 6 h. The mRNA levels were quantified by qRT-PCR and normalized to those of *EF1a* mRNA, while protein levels were determined by Western blot using the indicated antibodies, with tubulin used as the loading control. Protein band intensity was analyzed using ImageJ and normalized relative to tubulin. Results are reported as mean ± SEM (*n* = 3). * *p* < 0.05, ** *p* < 0.01 vs. control. The immunoblots shown are representative of at least three independent experiments.

**Figure 6 marinedrugs-21-00164-f006:**
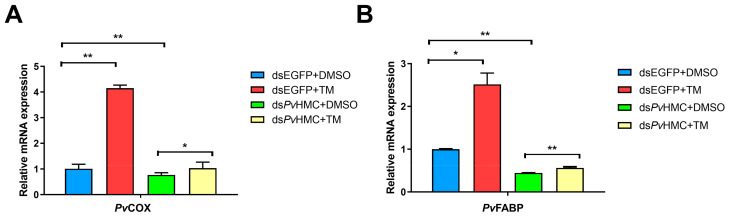
Hemocyanin mediates ER stress proteins to affect the expression of fatty acid metabolism genes in *Penaeus vannamei* hepatopancreas. mRNA levels of (**A**) *PvCOX* and (**B**) *PvFABP* after *Pv*HMC knockdown followed by treatment with 6 ng/µL of tunicamycin (TM) or diluted dimethyl sulfoxide (DMSO) for 6 h. The mRNA levels were quantified by qRT-PCR and normalized to those of *EF1a* mRNA. Results are reported as mean ± SEM (*n* = 3). * *p* < 0.05, ** *p* < 0.01 vs. control.

**Figure 7 marinedrugs-21-00164-f007:**
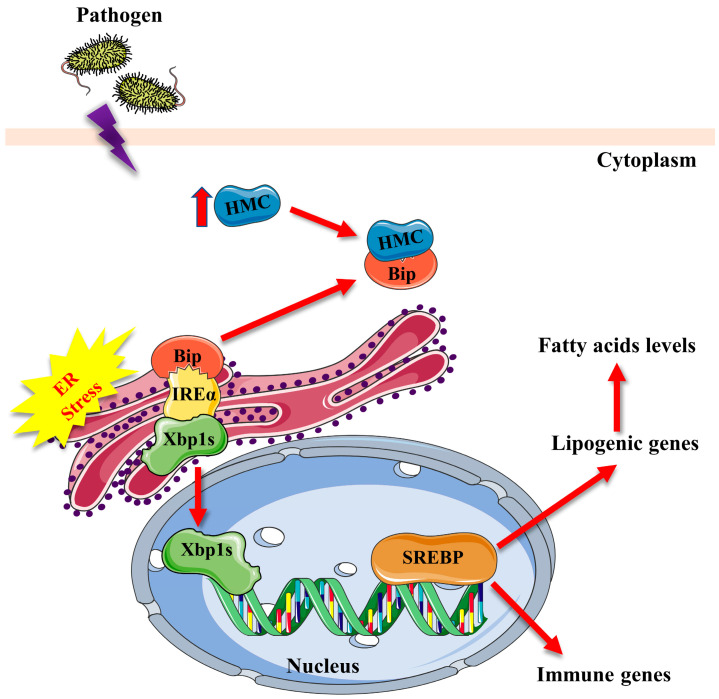
Diagrammatic illustration of a proposed mechanism by which hemocyanin (*Pv*HMC) mediates ER stress response to affect SREBP, lipogenic genes expression, and fatty acid levels in penaeid shrimp. Bacteria pathogens induce hemocyanin to activate ER stress by binding with ER stress proteins, such as Bip, which releases mature transcription factor *Pv*Xbp1s into the cytoplasm. *Pv*Xbp1s then enters the nucleus to induce the transcription and expression of *Pv*SREBP, resulting in the transcription of downstream target lipogenic genes, such as fatty acid binding protein (FABP) and cyclooxygenase (COX), to alter fatty acid levels, and also the expression of immune-related genes.

**Table 1 marinedrugs-21-00164-t001:** Fatty acid profile in hepatopancreas after *Vibrio parahaemolyticus* challenge.

Main Fatty Acids	Fatty Acid Composition (%) ^1^	
	Control (PBS)	*V. parahaemolyticus*
Palmitic acid (C16:0)	20.35 ± 0.58	21.64 ± 0.03
Palmitoleic acid (C16:1)	5.82 ± 0.20	6.82 ± 0.27
Stearic acid (C18:0)	0.11 ± 0.005	0.14 ± 0.002 *
Oleic acid (C18:1n-9c)	5.14 ± 0.07	5.27 ± 0.72
Linolelaidic acid (C18:2n-6c)	6.44 ± 0.20	6.59 ± 0.16
Arachidic acid (C20:0)	20.23 ± 1.51	16.37 ± 0.46
γ-Linolenic acid (C18:3n-6)	3.95 ± 0.18	4.05 ± 0.09
α-Linolenic acid (C18:3n-3)	2.04 ± 0.06	2.18 ± 0.007
Dihomo-γ-linolenic acid (C20:3n-6)	1.58 ± 0.06	1.49 ± 0.009
Arachidonic acid (C20:4n-6)	3.90 ± 0.05	4.70 ± 0.024 **
Eicosapentaenoic acid (C20:5n-3)	6.72 ± 0.11	7.90 ± 0.06 **
Docosapentaenoic acid (C22:5n-3)	0.87 ± 0.02	1.0 ± 0.003 *
Docosahexaenoic acid (C22:6n-3)	4.89 ± 0.16	5.18 ± 0.01
∑SFA	40.70 ± 2.07	38.16 ± 0.43
∑MUFA	10.96 ± 0.25	12.03 ± 0.20 *
∑LC-PUFA	30.40 ± 0.79	33.08 ± 0.07

^1^ Fatty acid composition was calculated as a percentage of total fat. * *p* < 0.05, ** *p* < 0.01.

**Table 2 marinedrugs-21-00164-t002:** Fatty acid profile in hepatopancreas after 4-Phenylbutyric acid treatment.

Main Fatty Acids	Fatty Acid Composition (%) ^1^	
	Control (PBS)	4-Phenylbutyric Acid (4-PBA)
Palmitic acid (C16:0)	11.03 ± 1.18	7.33 ± 0.95 *
Palmitoleic acid (C16:1)	1.44 ± 0.16	0.46 ± 0.06
Stearic acid (C18:0)	2.21 ± 0.17	2.02 ± 0.19
Oleic acid (C18:1n-9c)	11.93 ± 2.58	9.56 ± 1.39
Linolelaidic acid (C18:2n-6c)	4.31 ± 0.32	3.79 ± 0.37
Arachidic acid (C20:0)	16.09 ± 2.46	13.53 ± 1.53
γ-Linolenic acid (C18:3n-6)	0.1 ± 0.02	0.07 ± 0.008
α-Linolenic acid (C18:3n-3)	1.48 ± 0.29	0.97 ± 0.13
Dihomo-γ-linolenic acid (C20:3n-6)	1.07 ± 0.11	0.79 ± 0.09
Arachidonic acid (C20:4n-6)	2.01 ± 0.17	1.54 ± 0.07
Eicosapentaenoic acid (C20:5n-3)	2.03 ± 0.26	1.43 ± 0.39
Docosapentaenoic acid (C22:5n-3)	0.23 ± 0.03	0.09 ± 0.01 *
Docosahexaenoic acid (C22:6n-3)	1.45 ± 0.02	0.87 ± 0.005 **
∑SFA	29.33 ± 3.58	22.88 ± 2.67
∑MUFA	13.37 ± 2.72	10.01 ± 1.45
∑LC-PUFA	12.96 ± 0.96	9.57 ± 0.99

^1^ Fatty acid composition was calculated as a percentage of total fat. * *p* < 0.05, ** *p* < 0.01.

**Table 3 marinedrugs-21-00164-t003:** Fatty acid profile in hepatopancreas after tunicamycin treatment.

Main Fatty Acids	Fatty Acid Composition (%) ^1^	
	Control PBS	Tunicamycin (TM)
Palmitic acid (C16:0)	11.43 ± 0.03	13.23 ± 0.28 **
Palmitoleic acid (C16:1)	2.11 ± 0.25	1.06 ± 0.05
Stearic acid (C18:0)	2.06 ± 0.09	3.84 ± 0.22 **
Oleic acid (C18:1n-9c)	15.11 ± 0.83	14.75 ± 1.39
Linolelaidic acid (C18:2n-6c)	4.26 ± 0.10	6.22 ± 0.31 **
Arachidic acid (C20:0)	20.37 ± 0.64	22.59 ± 2.39
γ-Linolenic acid (C18:3n-6)	0.06 ± 0.003	0.16 ± 0.02 **
α-Linolenic acid (C18:3n-3)	2.17 ± 0.12	1.54 ± 0.11
Dihomo-γ-linolenic acid (C20:3n-6)	1.12 ± 0.12	1.37 ± 0.09
Arachidonic acid (C20:4n-6)	1.79 ± 0.16	3.32 ± 0.31 *
Eicosapentaenoic acid (C20:5n-3)	2.25 ± 0.17	4.14 ± 0.42 *
Docosapentaenoic acid (C22:5n-3)	0.23 ± 0.08	0.31 ± 0.11
Docosahexaenoic acid (C22:6n-3)	1.37 ± 0.17	2.06 ± 0.42
∑SFA	33.86 ± 0.69	39.66 ± 2.87
∑MUFA	17.22 ± 1.05	15.81 ± 1.34
∑LC-PUFA	13.25 ± 0.70	19.12 ± 1.15 *

^1^ Fatty acid composition was calculated as a percentage of total fat. * *p* < 0.05, ** *p* < 0.01.

**Table 4 marinedrugs-21-00164-t004:** Fatty acid profile in hepatopancreas after *Pv*HMC knockdown followed by tunicamycin treatment.

Main Fatty Acid	Fatty Acid Composition (%) ^1^			
	dsEGFP		ds*Pv*HMC	
	DMSO	Tunicamycin (TM)	DMSO	Tunicamycin (TM)
Palmitic acid (C16:0)	29.17 ± 0.27	33.49 ± 1.02 *	22.38 ± 0.33 **	22.45 ± 0.33
Palmitoleic acid(C16:1)	2.86 ± 0.03	3.87 ± 0.19 **	2.11 ± 0.56 **	2.17 ± 0.09
Stearic acid (C18:0)	5.64 ± 0.05	6.35 ± 0.24	5.12 ± 0.18	4.93 ± 0.64
Oleic acid (C18:1n-9c)	40.59 ± 0.45	45.57 ± 1.31 *	29.38 ± 0.57 **	28.99 ± 0.18
Linolelaidic acid (C18:2n-6c)	10.65 ± 0.34	12.39 ± 0.66	8.17 ± 0.12 **	8.08 ± 0.36
Arachidic acid (C20:0)	56.48 ± 0.80	62.41 ± 1.70 *	41.27 ± 0.80 **	40.57 ± 0.22
γ-Linolenic acid (C18:3n-6)	4.37 ± 0.30	5.29 ± 0.2 *	3.14 ± 0.06 **	3.18 ± 0.07
α-Linolenic acid (C18:3n-3)	2.69 ± 0.03	3.00 ± 0.04 **	2.01 ± 0.03 **	2.05 ± 0.01
Dihomo-γ-linolenic acid (C20:3n-6)	2.98 ± 0.05	3.18 ± 0.11	2.09 ± 0.04 **	2.01 ± 0.01
Arachidonic acid (C20:4n-6)	5.14 ± 0.03	5.84 ± 0.21 *	4.74 ± 0.05 **	4.93 ± 0.02 ^†^
Eicosapentaenoic acid (C20:5n-3)	5.46 ± 0.04	6.31 ± 0.19*	5.50 ± 0.06	6.01 ± 0.02 ^††^
Docosapentaenoic acid (C22:5n-3)	0.63 ± 0.01	0.73 ± 0.02 *	0.52 ± 0.005 **	0.52 ± 0.01
Docosahexaenoic acid (C22:6n-3)	6.74 ± 0.02	8.09 ± 0.23 **	6.08 ± 0.07 **	6.63 ± 0.03 ^††^
∑SFA	91.3 ± 1.12	102.25 ± 2.92 *	68.77 ± 1.02 **	67.95 ± 0.42
∑MUFA	43.45 ± 0.49	49.44 ± 1.51 *	31.49 ± 0.62 **	31.13 ± 0.27
∑LC-PUFA	38.65 ± 0.50	44.83 ± 1.60 *	32.27 ± 0.38 **	33.41 ± 0.48

^1^ Fatty acid composition was calculated as a percentage of total fat. * *p* < 0.05 and ** *p* < 0.01 compared with dsEGFP+DMSO, **^†^**
*p* < 0.05 and **^††^**
*p* < 0.01 compared with ds*Pv*HMC+DMSO.

## Data Availability

The data presented in this study are available on request from the corresponding author.

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
