# Peer review of "Modulation of SREBP Expression and Fatty Acid Levels by Bacteria-Induced ER Stress Is Mediated by Hemocyanin in Penaeid Shrimp"

_marinedrugs, 2023, doi:10.3390/md21030164_

Round 1

Reviewer 1 Report

Review comments

In this study entitled “Modulation of SREBP expression and fatty acid levels by bacteria-induced ER stress is mediated by hemocyanin in penaeid shrimp” by Huang et al, the authors reported that the expression levels of hemocyanin, ER stress proteins (Bip, Xbp1s, and Chop), SREBP, and fatty acids metabolism-related genes (FABP and COX) were induced in shrimp hepatopancreas by bacteria (Vibrio parahaemolyticus), together with changes in the fatty acids profile. Using transient transfections, RNAi-mediated knockdown, and inhibitor/activator treatment, they further revealed that hemocyanin plays a mediatory role in ER stress, which affected SREBP expression and fatty acids levels through an axis, which they named the "hemocyanin-ER stress-SREBP axis.” The study is well thought out, the experiments properly executed and the writeup and logical connections are straightforward and easy to follow. Moreover, the relevance of the study is vital in shrimp farming and aquaculture in general. That said, there are many things that need to be worked on in the manuscript to improve its quality before it can be accepted for publication. It is important to make these revisions to allow comprehension, especially by those who are not experts in this field.

The main areas that need revision are as follows:

1.      I suggest that authors should write the full names of SREBP and ER instead of the abbreviations

2.      Lines 14-15, I think the sentence “Many environmental and pathogenic insults induce endoplasmic reticulum (ER) stress in aquatic ecosystems where these factors are crucial for life” is a bit confusing. I think this should be revised clearly.

3.      Line 16, I suggest authors should add “expression” after “hemocyanin” in the sentence “…stressors induce hemocyanin”

4.      Line 17, “We demonstrate that in response to pathogenic bacteria…”, why not indicate the specific bacteria, or did authors demonstrate using all pathogenic bacteria?

5.      Line 59, please explain or define “HepG2 cells” for those who are not familiar with cell lines.

6.      Line 75, please define or state the full name of “Hsp70”

7.      Line 99, why did the authors limit their study to only pathogenic bacteria when in aquaculture, when viral infections, especially those caused by WSSV have had a huge impact on shrimp farming?

8.      Lines 106-107, why did the authors choose only V. parahaemolyticus and S. iniae for their study?

9.      Lines 109-110 or figure 1E, why did V. parahaemolyticus induce more of Bip protein (PvBip) expression than S. iniae? Can authors speculate on this?

10.  Line 177, and in tables 1-4, are these 13 fatty acids the only ones found in shrimp or the method used could only detect these fatty acids?

11.  Figures 4 and 6, table 4, DMSO can be toxic to cells and organisms, so how did authors control the toxic effect of DMSO on shrimp?

12.  Line 304 and throughout the manuscript, which isoform of hemocyanin was used/knocked down or overexpressed in this study, because per authors’ own previous studies (e.g., Zhang YL, et al. Front Immunol (2017) 8:611; Zhao S, et al. Immunol Lett (2013) 154:1–6) and studies by others (e.g., Xu, J. et al. Acta Oceanol. Sin. 34, 36–44 (2015); Johnson JG, et al. Integr Comp Biol (2016) 56:1080–91.), hemocyanin has several isoforms/subunits. Please clarify this in the manuscript.

13.  Lines 470-479, why did authors use both siRNA and dsRNA for their study? Can’t only one type be used?

14.  Line 530-531, for the in-house produced “Mouse anti-SREBP” and “Rabbit anti-hemocyanin” antibodies, how did authors ascertain the specificity of these antibodies?

Minor comments

1.      Line 356, “OSBPs” is duplicated so please delete one.

2.      Shrimp hepatopancreas contains many proteolytic enzymes, so how did authors ensure that these enzymes did adversely affect their samples.

3.      For the shrimps used in the study, are they SPF shrimp? if not how did the authors ensure that the shrimp didn’t contain other pathogens that would confound their data?

4.      In table 5, I suggest that the authors should include a column that shows the amplicon size.

Reviewer 2 Report

In this study, the authors found that bacteria regulate SREBP expression and fatty acid levels in shrimp through hemocyanin-mediated ER stress and proposed the hypothesis of a "hemocyanin-ER stress-SREBP axis" to account for this regulatory mechanism. These findings are interesting and provide an important insight into the regulation of SREBP expression and thus fatty acid metabolism by hemocyanin in response to pathogen-induced ER stress in crustaceans.

Major problems:

1. In this manuscript, authors found that hemocyanin, ER stress proteins (Bip, Xbp1s and Chop), SREBP and fatty acid metabolism-related genes (FABP and COX), and changes in fatty acid levels in the hepatopancreas are induced by pathogenic bacteria; they have a regulatory relationship with each other, but is this axial regulation a direct effect? HMC is able to bind to Bip, so why does it promote rather than inhibit the IREα and Xbo1s expression? Regarding the hypothesis "hemocyanin-ER stress-SREBP axis", many details need to be analyzed in more depth in order to give the reader a deeper understanding.

2. In the introduction, it is mentioned that ER stress leads to many cellular and physiological responses, so which one is the main one? Why is lipid metabolism emphasized here?

3. In the introduction, what are the specific consequences of ER stress leading to hepatic steatosis? Are there changes in fatty acid levels (lipids increase after bacteria-induced in this study) resulting in altered biological phenotypes?

4. What is the role of SREBP in immunity? it was not explained very clear in the introduction. Among the bacteria-induced changes in the expression of SREBPs, in what tissues do them most predominantly occur? Besides “hemocyanin-ER stress-SREBP”, are there any other more major regulatory ways?

5. Is hemocyanin the main molecule in the regulation of ER stress in the shrimp hepatopancreas, and are there any previous experiments or references?

6. There are many of hemocyanin genes in the shrimp P. vannamei (hundreds of sequences in NCBI), the specific gene selected here is X82502.1, why choose this gene? Is there any concern about the expression of other hemocyanin genes?

6. hemocyanin is a multi-copy gene, how to perform RNAi and whether it can be specifically knocked down?

7. in “2.2. Hemocyanin interacts with ER stress proteins”, no yeast two-hybrid experiment was mentioned, only in vitro GST pull-down analysis to prove the direct interaction between PvBip and PvHMC seems to be a little less evidence?

8. In the materials and methods, In the ER stress inhibitor or activator treatment experiments, treatment duration, muscle injection or hemolymph injection? After pathogenic challenge, was the hepatopancreas dissected for observation or bacterial culture experiments done? How do you confirm that the bacteria have attacked the hepatopancreas? Some details of the experiments need to be shown.

Round 2

Reviewer 2 Report

The author's response did a good job of clarifying some of the doubts in the previous version and I have no more questions.